# Inactivated Platelet Lysate Supports the Proliferation and Immunomodulant Characteristics of Mesenchymal Stromal Cells in GMP Culture Conditions

**DOI:** 10.3390/biomedicines8070220

**Published:** 2020-07-17

**Authors:** Katia Mareschi, Sara Castiglia, Aloe Adamini, Deborah Rustichelli, Elena Marini, Alessia Giovanna Santa Banche Niclot, Massimiliano Bergallo, Luciana Labanca, Ivana Ferrero, Franca Fagioli

**Affiliations:** 1Department of Public Health and Paediatrics, The University of Turin, Piazza Polonia 94, 10126 Torino, Italy; elena.marini@edu.unito.it (E.M.); alessiagiovannasanta.bancheniclot@unito.it (A.G.S.B.N); massimiliano.bergallo@unito.it (M.B.); franca.fagioli@unito.it (F.F.); 2Stem Cell Transplantation and Cellular Therapy Laboratory, Paediatric Onco-Haematology Division, Regina Margherita Children’s Hospital, City of Health and Science of Turin, 10126 Torino, Italy; scastiglia@cittadellasalute.to.it (S.C.); aadamini@cittadellasalute.to.it (A.A.); drustichelli@cittadellasalute.to.it (D.R.); iferrero@cittadellasalute.to.it (I.F.); 3Blood Component Production and Validation Center, City of Health and Science of Turin, S. Anna Hospital, 10126 Turin, Italy; llabanca@cittadellasalute.to.it

**Keywords:** mesenchymal stromal cells, good manufacturing practice, inactivated platelet lysate

## Abstract

Mesenchymal stromal cells (MSCs) isolated from bone marrow (BM-MSCs) are considered advanced therapy medicinal products (ATMPs) and need to be produced according to good manufacturing practice (GMP) in their clinical use. Human platelet lysate (HPL) is a good GMP-compliant alternative to animal serum, and we have demonstrated that after pathogen inactivation with psoralen, it was safer and more efficient to use psoralen in the production of MSCs following GMP guidelines. In this study, the MSCs cultivated in fetal bovine serum (FBS-MSC) or inactivated HPL (iHPL-MSC) were compared for their immunomodulatory properties. We studied the effects of MSCs on (1) the proliferation of total lymphocytes (Ly) and on naïve T Ly subsets induced to differentiate in Th1 versus Th2 Ly; (2) the immunophenotype of different T-cell subsets; (3) and the cytokine release to verify Th1, Th2, and Th17 polarization. These were analyzed by using an in vitro co-culture system. We observed that iHPL-MSCs showed the same immunomodulatory properties observed in the FBS-MSC co-cultures. Furthermore, a more efficient effect on the increase of naïve T- cells and in the Th1 cytokine release from iHPL was observed. This study confirms that iHPL, used as a medium supplement, may be considered a good alternative to FBS for a GMP-compliant MSC expansion, and also to preserve their immunomodulatory proprieties.

## 1. Introduction

Mesenchymal stromal cells (MSCs) are multipotent cells that can be isolated from a variety of tissues, capable of producing significant factors including multiple cytokines and growth factors. In recent years, their peculiar immunomodulatory characteristics, mediated by the release of a plethora of trophic factors in their secretome, have been considered as having more than their multilineage differentiation potential for their clinical use in severe disease clinical trials, namely: autoimmune, chronic inflammatory, and degenerative conditions [1,2].

MSCs isolated from bone marrow (BM-MSCs) are considered advanced therapy medicinal products (ATMPs) and need to be produced according to good manufacturing practice (GMP) [3,4] in their clinical use. As the use of xenogeneic protein-free GMP-compliant growth media is a prerequisite for clinical MSC isolation and expansion, human platelet lysate (HPL) has efficiently substituted fetal bovine serum (FBS) in MSC clinical manufacturing. For these reasons, it represents a good GMP-compliant alternative to animal serum for MSC clinical production, confirming recent data as reported in the literature [5,6,7]. As the risk of transmission of infective agents that have not yet been routinely tested, or for which no tests are currently available remains, HPL quality and safety has to be greatly improved. We have demonstrated that pathogen inactivation (PI) using psoralen was efficient in isolating and expanding safer MSCs under GMP conditions [8]. PI technology was used for the first time in 1991 [9] to treat fresh-frozen plasma, and then later for platelets and red cells (RBCs) [10]. This technology efficiently removes a wide range of pathogens, with no toxicity or counter effect on the product potency, and may furthermore prevent the transmission of unknown pathogens [11,12,13,14,15]. For this reason, to prepare HPL more safely for MSC GMP production, we used pathogen inactivation using psoralen and following GMP guidelines [8]. We also demonstrated that HPL subject to pathogen inactivation (inactivated HPL (iHPL) was more advantageous in MSCs isolated from bone marrow (BM-MSCs) in terms of their cellular growth and stemness. On the basis of our findings for iHPL and the literature data on a possible decrease of immunomodulatory properties of MSCs cultured in HPL [16], we therefore studied various iHPL effects on the immunomodulatory properties of MSCs.

In our study, BM-MSCs were isolated and expanded simultaneously in iHPL (iHPL-MSCs) in GMP compliant conditions and also in FBS (FBS-MSCs) under standard conditions, and as commonly reported in immunomodulatory studies [17]. We focused particularly on the effects of FBS-MSCs and iHPL-MSCs on T lymphocytes (Ly), as investigated in a previous paper [18]. MSCs do not express MHC class II and costimulatory molecules, such as CD40, CD80 or CD86, moreover, studies show that MSCs are able to inhibit or limit inflammatory responses and mitigate anti-inflammatory pathways, thus directly or indirectly inhibiting disease-associated Th1, Th2, and Th17 cells, as well as cytotoxic T lymphocytes [17,18,19,20]. The effects of MSCs cultivated with the two different supplements were analyzed by using an in vitro co-culture system with peripheral blood mononuclear cells (PBMCs) stimulated with phytohemagglutinin (PHA-PBMC). The following effects were analyzed:the effect on the proliferation of the total Lythe effect on the proliferation of naïve T Ly subsets induced to Th1 and Th2 Ly differentiationthe immunophenotype of different T cell subsets (naïve, memory, effector, and Th1 and Th2 lymphocytes)

## 2. Experimental Section

### 2.1. Human MSC Isolation, Expansion, Analysis, and Characterization

Human BM samples were collected from healthy adult donors (mean age and SEM: 26 ± 6). The discarded collection bag following the filtration of bone marrow (BM) was washed with phosphate buffer saline (PBS) 1× (Lonza, Versviers, Belgium) and the cells were collected. The donors signed an informed written consent in accordance with the Declaration of Helsinki and the City of Health and Science of Turin—Ordine Mauriziano Hospital Ethics Committee. which approved the BM collection from donors for research use on May 16, 2016. Then, they underwent BM collection for a familiar allogeneic hematopoietic stem cell transplantation for pediatric patients with severe leukemia treated at our center—Regina Margherita Children Hospital, Pediatric Onco-Hematology and Stem Cell Transplant Division.

The BM cells were counted, and were simultaneously plated in the two FBS and iHPL conditions containing α MEM (Biochrome, Berlin, Germany), 2mM L-glutamine (Sigma-Aldrich^®,^ St. Louis, MO, USA.), and penicillin/streptomycin 1× (Euroclone, Pero, Mi, Italy) supplemented with 10% of FBS (Sigma-Aldrich^®,^ St. Louis, MO, USA) or 10% of iHPL in T75 o T150 flasks (Becton Dickinson, Franklin Lakes, NJ, USA) at 1 × 10^4^ cells/cm^2^. The HPL was prepared from a platelet pool of healthy donors and underwent pathogen inactivation by psoralen, as described in the literature [8], at the Blood Component Production and Validation Center, City of Health and Science of Turin, S. Anna Hospital.

The non-adherent cells were removed after five to seven days of culture, and the adherent cells were refed every three to four days, detached with trypsin/EDTA 1× (Sigma-Aldrich^®^) for 5 min at 37 °C when they reached semi-confluency, and expanded for several steps until they no longer reached confluence.

The cells were analyzed following the International Cellular Society MSC criteria [21] and frozen in FBS with 10% dimethyl sulfoxide (DMSO, Euroclone, Pero, Mi, Italy), or in a physiological solution containing 5% human albumin and 10% of DMSO. For this study, we thawed five BM-MSC batches at the second or third passage, and analyzed their viability, immunophenotyped, and differential proliferative potential to verify whether the freezing process had preserved their characteristics [18]. The isolated BM-MSCs, expanded in FBS or in iHPL, were referred to as FBS-MSCs and iHPL-MSCs, respectively.

### 2.2. Preparation of Human Peripheral Blood Mononuclear Cells (PBMC)

The PBMC were separated from buffy coats by density gradient centrifugation using Ficoll Histopaque (Sigma-Aldrich^®^). The buffy coats were collected from five healthy adult donors aged between 18 and 65 years old, following their informed consent and using an automated blood component separator (Compomat G5, Fresenius Kabi; Bad Homburg, Germany) at the Blood Component Production and Validation Center, City of Health and Science of Turin, S. Anna Hospital. The donors were negative for infection markers (Human immunodeficiency virus 1–2, Hepatitis B and C, and Treponema pallidum), in accordance with Italian law and European guidelines.

### 2.3. Co-Culture MSCs/T Cells

We performed all of the co-culture experiments following the same experimental design as described in the literature [18], namely: BM-MSCs were allowed to adhere overnight and were incubated at 37 °C and 5% CO_2_. The medium was then removed and the total PBMCs from an unrelated donor were added. The experiments were performed with five MSC batches and five unrelated PBMC donor pairs. We also used CD45RA microbeads (Miltenyi Biotech, Bergisch Gladbach, Germany) and followed the manufacturer’s instructions to isolate the T naïve cells. To trigger the T lymphocytes, phytohemagglutinin (PHA; 2.5 μg/mL) was added in the PBMCs. Then, the PBMCs were stimulated with anti-CD3, anti-CD28, rh- Interleukin (IL)-2, rhIL-12, and anti-IL-4 to trigger the Th1 cells, and anti-CD28, anti-CD3, rhIL-4, rhIL-2, and anti-Interferon (IFN)-γ to trigger the Th2 cells, as described in the literature [18].

The culture conditions were as follows: (1) unstimulated PBMC, (2) MSCs alone, (3) PBMC stimulated with PHA (PHA-PBMC), (4) MSCs with unstimulated PBMC, (5) MSCs with PHA-PBMC, (6) Th1 or Th2 induced CD45RA+, or (7) MSCs with Th1 or Th2 induced CD45RA+.

Five days later, the non-adherent cells in the co-cultures were harvested and counted for cytofluorimetric and proliferative analyses.

For the proliferation test, 2 × 10^3^ cells/well of MSCs were irradiated at 3000 rad and plated into 96-well plates in 100 μL of complete α-MEM medium in triplicates. Then, 24 h later, 2 × 10^4^ PBMCs with PHA were added to the wells with or without MSCs. The MSCs/PBMC ratio was 1:10. The condition without PHA was also used as a control. The cells were harvested after 72-h of co-culture and after 4 h of ^3^H-thymidine (1 μCi (0.037 MBq) incubation, and the incorporated radioactivity was evaluated in counts per minute (cpm) using a 1450 Microbeta TriLux apparatus (Perkin Elmer, Boston, MA, USA).

### 2.4. Cytofluorimentric Analysis

All of the antibodies (mAb) for PBMC characterization were used with the appropriate amount of antibody, following the antibody titration test described by Rustichelli et al. [22] and following this combination: anti-human CD45RA-FITC/CD45RO-PE/CD3-PercP-Cy5.5/CD4-APC, CD45RA-FITC/CD45RO-PE/CD3-peridinin-chlorophyll protein cyanine 5.5 (PerCP-Cy5.5)/CD8-APC. The labeled cells were acquired on a FACSCanto II (Becton DickinsonBrea, CA, USA) using the DIVA software program after washing with PBS 1× (200× *g* for 10 min) and multiparameter acquisition list modes were analyzed by Kaluza Analysis Software 2.1 (Becton Coulter, Brea, CA, USA).

The unstained cells were used as a negative control, and the percentage of positive cells was used to calculate the absolute number on the basis of the cell number counted following five days of co-culture.

### 2.5. Evaluation of Cytokine Release by ELISA

We evaluated the Th1, Th2, and Th17 cytokine release (IL-2, IL-12, IFN-γ, Tumor necrosis factor (TNF)-α, IL-10, IL-17, IL-4, and IL-6) on the collected supernatants of all of the culture conditions using a coated Enzyme Linked Immunosorbent Assay ( ELISA) kit (Mabtech, ELISA Assay, AB, Nacka Strand, Sweden), as previously described in the literature [23].

### 2.6. Statistical Analysis

All of the data obtained in this study were analyzed using Graph PAD Prism (version 8.0.0, for Windows, GraphPad Software, San Diego, California USA, www.graphpad.com) statistical software. All of the mean values with a standard error of mean (SEM) were reported in the graphs. The effects of the FBS-MSCs and iHPL-MSCs on the T lymphocytes were analyzed in the experiments with repeated measures using two-way ANOVA. A multiple comparison Tukey test was used to compare each mean, and a Dunnett test was used to compare the co-culture conditions with the control condition (PHA-PBMC).

All of the statistical tests were considered significant for *p* < 0.05, highly significant for *p* < 0.001, and very highly significant for *p* < 0.0001.

## 3. Results

### 3.1. MSC Characteristics

The Thawed FBS-MSCs and iHPL-MSCs grew and reached confluence within a few days. Prior to use, the cells were analyzed for immunophenotype and multipotent characteristics, as demonstrated in the literature [8]. The MSCs, independently from the culture condition, were negative for CD45, CD34, and CD14 (0.70 ± 0.26 and 0.56 ± 0.18 for FBS-MSCs and iHPL-MSCs, respectively) and were positive (over 95%) for CD90 (a membrane glycoprotein, also called Thy-1), CD105 (endoglin), and CD73. CD146 (cell surface glycoprotein MUC18) was also tested, and was positive in all of the samples and was 78.33 ± 7.2 for FBS-MSCs and 75.3 ± 9.2 for iHPL-MSCs. No statistical differences were observed between the two groups in terms of both positive cell percentages and fluorescence means of the positive markers, as shown in Figure 1.

### 3.2. MSC/T Cell Interaction and Proliferative Assay

The 3H-thymidine incorporation data from each experiment was expressed as the mean of counts per minute (cpm) of the triplicate, as can be seen in the graph in Figure 2, with their SEM. PHA-PBMC showed high values of cpm (mean with SEM: 140,265 ± 34,868), which significantly decreased in the co-culture conditions, namely, 33,911 ± 6919 in FBS-MSCs and 43,723 ± 9339 in iHPL-MSCs, respectively. Two ANOVA tests and multiple comparison analyses showed a significant difference between PHA-PBMC versus PHA-PBMC + FBS-MSCs (*p* = 0.007), and between PHA-PBMC versus PHA-PBMC + iHPL-MSCs (*p* = 0.0016), whereas no significant differences were observed between PHA-PBMC + FBS-MSCs versus PHA-PBMC + iHPL-MSCs.

Variant modulations in the proliferative activity in Th1 and Th2 induced PBMC were observed, but without significant differences between FBS-MSCs and iHPL-MSCs.

### 3.3. T Cell Subsets Determination

The multiparameter flow cytometric analysis allowed for the identification of the following T subsets, based on the antibody combination used:Cytotoxic T naïve cells: CD45RA^+^/CD3^+^/CD8^+^Th naïve cells: CD45RA^+^/CD3^+^/CD4^+^Cytotoxic T memory cells CD45RO^+^/CD3^+^/CD8^+^Th memory cells: CD45RO^+^/CD3^+^/CD4^+^

From the percentage obtained from the cytofluorimetric analysis, we calculated the absolute number of cells after five days of co-culture (PHA-PBMC + FBS-MSCs and PHA-PBMC + iHPL-MSCs). The data, obtained for naïve and memory T subsets from 10 experiments, are summarized in Figure 3. The percentage of these subsets was strictly related to the variability of the donors. Despite obtaining a variable number in the PHA-PBMC, we observed a strong statistical significance in both the co-cultures with FBS-MSCs and iHPL-MSCs in comparison with the single culture of PHA-PBMC. Interestingly, in all of the experiments, the different subsets showed the same modulation trend. In particular, in PHA-PBMC, the memory T-cells were higher than the naïve cells, and, after co-culture with both FBS-MSCs and iHPL-MSCs, it was observed that (a) this ratio reversed in favour of naïve T cells, especially naïve cytotoxic subsets, and (b) both CD4 and CD8 memory T cell subsets significantly decreased. The two-way ANOAVA multiple comparison showed a significant increase in naïve cells CD45RA+, both helper (CD45RA+3+4+) and cytotoxic (CD45RA+3+8+), in co-cultures with PHA-PBMC + iHPL-MSCs with *p* = 0.0302 and 0.1076, respectively, when compared to the PHA-PBMC. The decrease of CD45RO+3+4 and CD45RO+3+8+ memory cells was highly significant in all of the co-culture conditions, as shown in Figure 3. In particular, when comparing PHA-PBMC alone versus PHA-PBMC + FBS-MSCs, and versus PHA-PBMC + iHPL-MSCs, we observed a *p* = 0.002 and *p* = 0.007 for CD45RO+3+4+, and *p* = 0.0010 and 0.0017 for CD45RO+3+4+, respectively. However, there were no significant differences in the co-cultures for the FBS-MSC and iHPL-MSC groups.

### 3.4. Th1/Th2/Th17 Cytokine Release

The PHA-PBMCs were used as a control group, and the range of analysed cytokines is reported in Figure 4. As also described in the literature [23], PHA-PBMC showed high levels of IL-2, IL-12, TNF-α, and IFN-γ, and in all of the co-culture conditions we observed a strong decrease in these cytokine releases. We also analysed the constitutive expression of these in cytokines in FBS and iHPL-MSCs, and we observed no difference between them in any of the following cases: (a) negligible levels of Th1 cytokine, except for IL-12, which was higher than the IL-12 levels in PHA-PBMC; (b) very low Th2 cytokines; and (3) significant levels of IL-6 and IL-17. For all of the cytokines, the multiple comparison statistical analysis did not reveal any significant differences between FBS-MSCs and iHPL-MSCs. In this case, we focused our statistical analyses on the effects of the different MSCs on PHA-PBMC using the Dunnett’s test.

In the co-culture experiments, we observed a significant decrease in the IL-2 concentration both in FBS-MSCs (*p* = 0.0112) and with iHPL-MSCs (*p* = 0.0007). However, IL-12 was produced in high levels as the PHA-PBMC decreased, even if FBS- MSCs and iHPL-MSCs constitutively produced higher levels of this cytokine than PHA-PBMC. We thus observed a reduction of IL-12 in the two co-culture conditions, with a statistically significant decrease in the co-culture with iHPL-MSCs (Figure 4).

The high level of TNF-α produced by PHA-PBMC decreased in all of the MSC co-culture conditions, as shown in Figure 4C. IFN-γ was also significantly decreased in the co-cultures with MSCs (Figure 4).

The PHA-PBMCs produced moderate amounts of Th2 cytokine levels (IL-4 and IL-10), which increased in the presence of MSCs in all of the co-culture conditions, but without statistically significant differences in the two co-culture conditions (Figure 4).

Moreover, PHA-PBMC showed a moderate production of IL-17 and IL-6, which increased in all of the co-culture experiments with MSCs. The data are reported in Figure 4, and multiple comparison tests showed a significant increase of IL-6 in the co-cultures with FBS-MSCs and iHPL-MSCs (*p* = 0.0402 and *p* = 0.0469, respectively), but no statistical differences were observed in the IL-17 levels. Furthermore, no differences were observed between the FBS-MSCs and iHPL-MSCs.

## 4. Discussion

Regenerative medicine is of growing interest in biomedical research, and in this context, MSCs are a promising tool for cell therapies because of their multipotent, bystander, and immunomodulant proprieties. For these reasons, MSCs are used for a very wide range of therapeutic applications, the majority of which are in phase I, phase II, or a mixture of phase I/II studies. Some phase III and IV studies are also in progress [24]. A variety of protocols are described for GMP MSC production, some of these using selected FBS, others using xeno-free components, HPL, or plasma. As MSCs are considered ATMPs, qualified protocols with standardised characteristics are needed so that MSCs can be developed in large-scale quality and at a relatively low-cost of production using xeno-free media for clinical-grade expansion [25,26]. In recent years, we have been establishing methods to isolate and expand MSCs for clinical use, and we have also demonstrated that iHPLs prepared in-house from a large pool of donor platelets, that undergo pathogen inactivation using psoralen, are safer and more efficient than FBS in isolating BM-MSCs when following GMP guidelines [8,27]. It has emerged from the analyses of the pluripotency markers, such as Oct-3/4 and NANOG, that the proliferation and differentiation properties of these iHPL-MSCs might be linked to a more immature stemness compared with FBS-MSCs. These observations indicate that iHPL-MSCs contain a subpopulation of multipotent stem cells, which might be the precursors of MSCs, with a more primitive phenotype than those of FBS-MSCs [8].

In this study, we wanted to know whether the MSCs that were isolated and expanded in iHPL preserved their immunomodulant proprieties by analysing their effects on T lymphocytes, and comparing them with the MSCs that were isolated and expanded in FBS with immunomodulating properties that have been described in the literature [17,18,28].

The ability to modulate the alloreactive immune response has been documented for MSCs derived from human BM; concurrently comparative studies between FBS-MSCs and HPL-MSCs have already been done and have indicated that HPL, used as a supplement for MSC media, supports immune modulation, at least to the same extent as FBS [29,30,31]. No comparative studies of immunomodulation were done to observe the differences between FBS-MSCs and the MSCs cultured in iHPL, which are even safer and more GMP compliant than HPL for MSC expansion [8].

BM-MSCs cultured in alpha-MEM + 10% FBS or 10% iHPL (FBS-MSC and iHPL-MSC) aliquots were thawed and, following immunophenotypic analysis to confirm their MSC identity (Figure 1), were analysed for their immunomodulatory effects on the T-cells of healthy donors. As previously demonstrated, the results, in accordance with the literature, showed that MSCs cultured in both FBS and iHPL are multipotent stem cells with immunophenotypic characteristics and differentiation potential, as established by the Cellular Therapy Society guidelines [21].

We evaluated and compared the inhibitory effects of MSCs on the total activated T-cells stimulated with a potent mitogen (PHA) and Th1 and Th2 effector cells generated from induced naïve T-cells. We observed a strong inhibitory effect of both MSCs on PHA-PMBC proliferation. The proliferation data obtained in the induced Th1 and Th2 effector cells showed variant modulations and no statistically significant differences. However, T-cell inhibition might be related to the interaction of T cells with dendritic cells and Natural Killer (NK) which were not present in Th1- and Th2-induced co-cultures.

We observed that FBS-MSCs and iHPL-MSCs inhibited T-cell proliferation without significant differences. In the co-cultures of T-lymphocytes with iHPL-MSCs or with FBS-MSCs, we analysed the different T subsets, and we observed a statistically significant increase of naïve T-cells in the condition with iHPL-MSCs, as well as a strong decrease of memory T-cells in both co-cultures with iHPL-MSCs and FBS-MSCs. Indeed, the presence of iHPL induced a significant increase of CD4+ and CD8+ naïve T-cell subpopulations compared with stimulated PHA-PBMC, showing higher effects of naïve T-cells than FBS in the co-culture with MSCs. The effect of MSCs on the increase of naïve T cells and the decrease of cytoxic T-cells has already been reported in the literature [32,33], as well as in several clinical trials demonstrating that the administration of MSCs reduced graft versus host disease (GVHD) and ameliorates different autoimmune and/or inflammatory diseases, where the T-cell functional compartment is involved in their pathophysiology [34,35].

The paracrine action of cytokines produced from MSCs represents an important mechanism that induces beneficial effects in cell therapy. In this study, pro-inflammatory and anti-inflammatory major cytokines were analysed and compared. We detected high quantities of IL-12 and negligible amounts of TNF-α, IL-2, IFN-γ, and IL-10 in MSCs, as also reported in the literature [36,37]. We also observed high levels of Th1 cytokine in the PHA-PBMC condition, as reported by Lee et al. [23].

In all of the co-culture experiments, the Th1 cytokines significantly decreased. The interaction MSCs/T-cells might block the Th1 polarization, because this phenomenon was also found for IL-12 produced in high concentrations, also by MSCs. We also noted that the major anti-inflammatory cytokines (IL-4 and IL-10) increased in co-cultures with FBS-MSCs and iHPL-MSCs, but showed no statistical differences. Our results indicate that FBS-MSCs and iHPL-MSCs induce a change from the cytokine secretion profile, from an inflammatory one to an anti-inflammatory, as broadly reported [2,37,38].

The same effect was observed for the Th17 cytokines. Moreover, iHPL- and FBS-MSCs produced high levels of both IL-6 and IL-17. We observed a statistically significant increase of IL-6 in co-cultures with MSCs in comparison with PHA-PBMC, while it is difficult to account for the significant increase of IL-17 in the co-culture with PHA-PBMC.

In this study, we confirmed the literature data, because the MSCs in the two experimental conditions inhibit or limit inflammatory responses, promoting the mitigation of anti-inflammatory pathways, especially in Th1 cells, and inducing a paradoxical increase in Th17 cells. A mechanism that could explain this increase of pro-inflammatory Th17 cells is the high quantity of IL-6, which is also a mediator of Th17 cell differentiation [28,39,40]. Furthermore, we noted that MSCs induced an upregulation of IL-6 inhibiting Th1 subset differentiation and an upregulation of IL-4 and IL-10, probably produced by a cellular compartment different from the T-cells inducing Th2-deviated immune response. However, it is difficult to draw conclusions on the separate roles of the different cytokines in either directly or indirectly mediating inhibition, because of the complex interaction of many factors.

On the basis of our results, we observed that iHPLs preserve all of the analysed immunomodulant proprieties of MSCs and FBS. For this reason, they may be more effective than FBS-MSCs in the control of immune-mediated pathologies, particularly in the case of GVHD. On the other hand HPL-MSCs appeared to be particularly useful in regenerative medicine [16].

The advantage of HPL in obtaining MSCs in vitro expansion is related not only to the increment of proliferation, but also to the fact that, differently from FBS, HPL does not induce antibody responses in patients [41,42], and its inactivation with psoralen makes it even safer than HPL by blocking and inactivating the replication of viruses, bacteria, and leukocytes in PLT concentrates [43]. The fact that we, and others [8,11], found differences in the immunomodulant properties of FBS-MSCs and HPL/iHPL-MSCs, which, in our case, are not significant, offer an interesting clue to the possible functional differences in MSC output and their clinical application. It is of note that we prepared HPL batches from 60 different donors, to standardize the medium supplement in both a growth factor concentration and in inter-donor variability before pathogen inactivation, which makes it more secure/safer for MSC GMP manufacturing.

In conclusion, as iHPL shows a greater proliferation, differentiation, and stemness potential in MSCs than FBS [8] and preserves the immunomodulant properties of BM-MSCs, a more efficient additive alternative to FBS for a GMP-compliant MSC expansion may be considered on T-cells.

## 5. Conclusions

The development of new strategies for the large-scale production of these cells, according to current regulations, including GMP, represents a fundamental step towards allowing their use in effective therapeutic approaches.

The use of iHPL as an alternative to FBS to isolate and expand MSCs confirmed that it is possible to obtain a large number of MSCs for clinical doses, which keep all of their characteristics intact, including their immunomodulatory properties. The application of iHPL as a medium supplement further reduces the manufacturing time, limiting the number of steps and reducing the starting volume of BM. It was also found that the pathogen inactivation treatment did not modify the characteristics of HPL, and made it safer and more suitable for MSC isolation and expansion for clinical use, and therefore, could be a requirement for GMP MSC expansion.

This study may well open the way for new ambitious projects, which will allow for the identification of new and more advantageous ways to culture MSCs, much safer than those commonly used, in order to create a bio-bank of ready-to-use MSCs for clinical use. The possibility of banking MSCs isolated from BM or other sources under GMP conditions in an AIFA (Agenzia Italiana del Farmaco—the Italian Medicine Agency) accredited cell factory might represent a new scenario for their clinical use in cell therapy protocols, providing for a continuous supply of cells to treat patients with acute GVHD after allogeneic hematopoietic stem-cell transplantation, solid organ transplantation, or in inflammatory and autoimmune diseases.

## Figures and Tables

**Figure 1 biomedicines-08-00220-f001:**
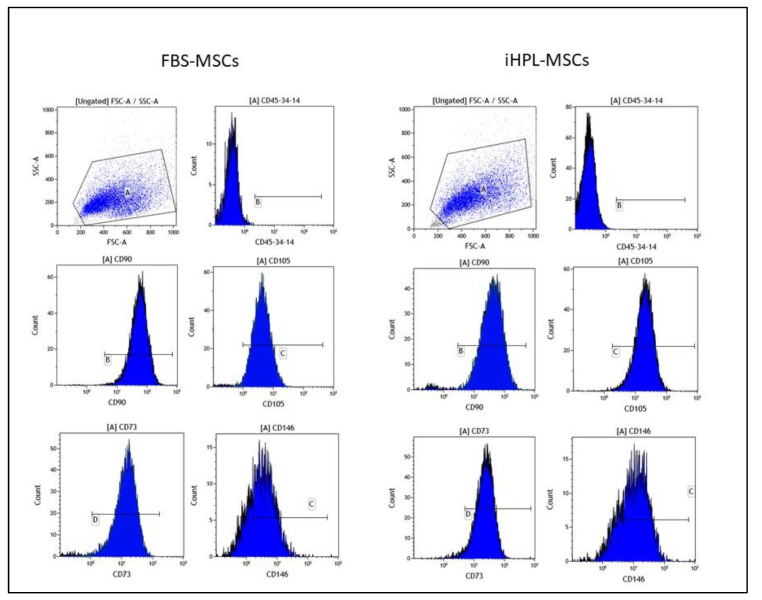
Immunophenotype of a representative cytofluorimetric analysis of fetal bovine serum mesenchymal stromal cells (FBS-MSCs) and inactivated human platelet lysate (iHPL)-MSCs cultures post thawing.

**Figure 2 biomedicines-08-00220-f002:**
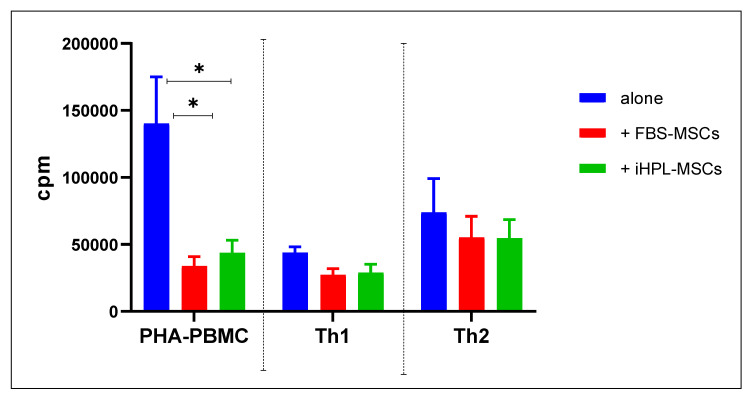
Proliferative assay on stimulated peripheral blood mononuclear cells (PBMCs) and on induced Th1 and Th2 cells alone, and in co-culture with FBS-MSCs or iHPL-MSCs. Each histogram represents the mean values ± standard error mean (SEM) of each condition (*n* = 5), and the symbol * indicates statistically significant differences with *p* < 0.05.

**Figure 3 biomedicines-08-00220-f003:**
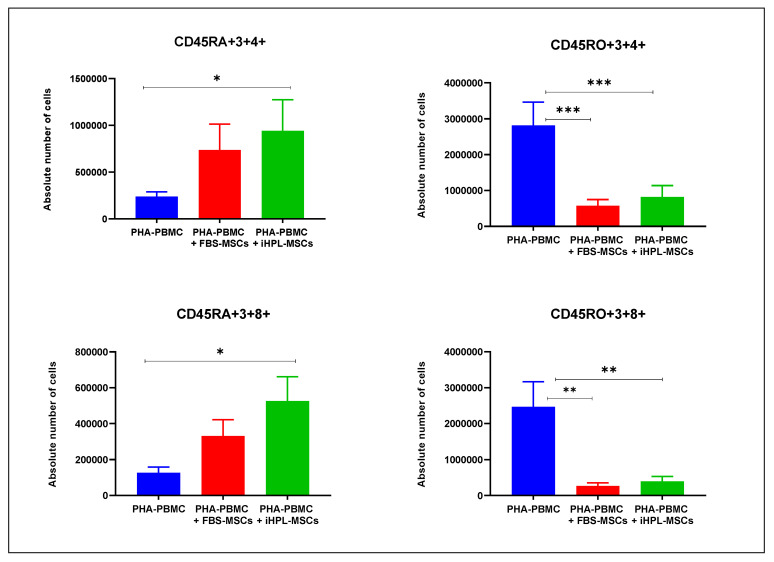
T subsets evaluated in phytohemagglutinin (PHA)-PBMC alone and in co-cultured conditions with FBS-MSCs and iHPL-MSCs, and expressed in mean values ± SEM. The symbols *, **, and *** indicate statistically significant differences with *p* < 0.05, *p* < 0.01, and *p* < 0.001, respectively.

**Figure 4 biomedicines-08-00220-f004:**
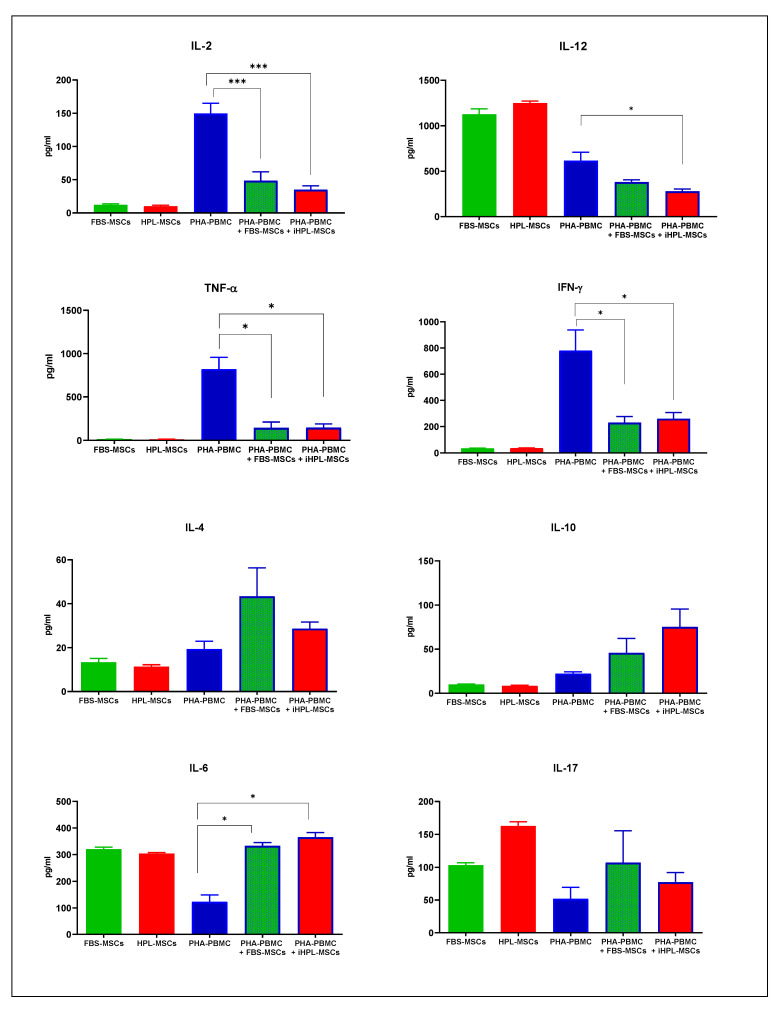
Th1 (IL-2, IL-12, TNF-α, and IFN- γ), Th2 (IL-4 and IL-10), and Th17 (IL-6 and IL-17) cytokine release (pg/ml) analysed using ELISA assay in FBS-MSCs, HPL-MSCs, and PHA-PBMC in isolation, and in the co-cultures PHA-PBMC with FBS-MSCs or with iHPL-MSCs, expresses with mean values ± SEM (*n* = 3). The symbols * and *** indicate statistically significant differences with *p* < 0.05 and *p* < 0.001, respectively.

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
