# Peer review of "Inactivated Platelet Lysate Supports the Proliferation and Immunomodulant Characteristics of Mesenchymal Stromal Cells in GMP Culture Conditions"

_biomedicines, 2020, doi:10.3390/biomedicines8070220_

Round 1
Reviewer 1 Report
This study compared bone marrow-derived MSCs that were isolated and expanded simultaneously in media containing either iHPL that was prepared using Psoralen as a pathogen inactivator or FBS. The immunophenotype of MSCs was measured. The effect of the two different MSCs were also compared in co-culture experiments with PHA-PBMCs and the different T cell subsets (naïve, memory, effector, Th1 and Th2 lymphocytes were assessed after 5 days of culture. The proliferation of PHA-PBMCs, Th1 and Th2 cells as well as the Th1 and Th2 cytokine release profile was additionally measured. They showed that iHPL-MSCs (Psoralen) had similar properties to MSCs cultured in FBS and that there were no significant differences between the two groups suggesting that iHPL-MSCs (Psoralen) could be used for GMP production of MSCs.
Major comments:
- Table I should be removed. It is inaccurate and suggests something different than what the data actually shows. Moreover, the authors state that "for each experiment a contingency table is given (Table 1- Summary of results)". This suggests that the Table is a summary of the results of the present study. The table also makes the reader believe that MSCs cultured in iHPL have a better proliferation potential which was not shown in the present study, despite being shown in reference (8). Additionally, pluripotency was not measured in the present study. And importantly, as the present study clearly showed, there were no significant differences between FBS and iHPL co-cultures. Therefore Table I is inaccurate and gives a biased impression of the results.
- Methods are insufficient and need to provide more details rather than refer to other papers. Some examples. A) Specify which antibodies were used for FACS analysis. B) Define what the Physiological solution is pg. 3 and from which company, as well as where they purchased human albumin, PHA, CD45RA microbeads. C) Define which cocktail of antibodies and cytokines were used for Th1 vs. Th2 polarization, company and more details. D) At which point were the cells frozen specifically? E) Include the passage at which the cells were frozen and used for experiments as this could influence results.
- In the Methods they state that MSCs were expanded for several passages until they no longer reached confluence. They should specify how long it took until cells no longer reached confluence as this is important. Was there a difference between FBS and iHPL. Could they include this as experimental data?
- The authors suggest that some of the cells were not compliant with ICS MSC criteria which is highly unusual and that these cells were not used. More specific details need to be provided on their selection criteria. What was their selection criteria and how specifically did they choose "compliant" cells vs. non-compliant cells?
- Were all MSCs completely negative for CD45, CD34 and CD14? The mean +/- SD should be provided. The authors should state that the CD45, CD34 and CD14 data was not shown (data not shown).
- The % positive CD146 cells should be provided especially since HPL has been shown to influence the expression of CD146 and this has been correlated with altered bone marrow MSC and platelet MSC differentiation properties (Choice of Xenogenic-Free Expansion Media Significantly Influences the Myogenic Differentiation Potential of Human Bone Marrow-Derived Mesenchymal Stromal Cells doi: 10.1016/j.jcyt.2015.11.019; Expression of CD146 on Human Placenta-Derived Mesenchymal Stromal Cells and their Osteogenic Differentiation Capacity are Modulated by Factors Contained in Platelet Lysate, doi: 10.4172/2325-9620.1000133).
- In lines 33-35 in the Discussion on page 10, the authors state that "Proliferative and differentiative characteristics were not modified (data not shown) by cryopreservation". If the authors have the data, they should consider showing this as this would enhance the paper especially since several groups have shown differences in differentiation potential in FBS vs xenogenic-free cultures, while others have shown no differences. It would be interesting to include the data since they used a different method of HPL inactivation (Psoralen) which is not commonly used.
- "MSCs also produced a moderate concentration of TNF-α". Is this in response to culture? Have others shown similar results in levels of cytokines measured under similar conditions using FBS or HPL (this should be discussed)? There is one citation for PHA-PBMCs but what about MSC studies? Also, what was the health and age of the PBMC and MSC donors (include in methods) as this could lead to e.g. high production of TNFa.
- Also, in lines 21-24 in 3.3 Results, they state that “ In particular, in PHA-PBMCs memory T cells were higher than naïve cells and after co-culture with both FBS-MSCs and iHPL-MSCs it was observed that: a) This ratio reversed in favour of naïve T cells, especially naïve cytotoxic subsets”. This is interesting. What does this indicate? The authors should speculate on this in the Discussion.
- Lines 39-46 Discussion paragraph need to be adapted and their data should be compared to reference (8). Moreover, this study did not "... confirm that iHPL, used as medium supplement, may be considered a more efficient additive alternative to FBS for a GMP-compliant MSC expansion" and the sentence/paragraph should be modified to reflect the data as no significant differences were shown between the two groups.
- English needs work. It would be beneficial to have an English speaker read through this and correct it as there are multiple areas that need modifications.
Minor comments:
- Abstract – MSCs should not be capitalized and the comma should be removed after MSCs. There is a lot of incorrect use of commas.
- BM – abbreviation not defined in methods
- Line 36 Discussion, change abbreviation of FBS-MSsC to FBS-MSCs
- In lines 25-31 in 3.3 Results, the authors need to state which comparisons were done to obtain these statistical values (e.g. x compared to z group). For example, "A Two Way Anova comparison showed a significant increase of naïve cells CD45RA both helper (CD45RA+3+4+) and cytotoxic (CD45RA+3+8+) in co-cultures with PHA-PBMC 26 + iHPL-MSCs respectively with p=0.0302 and 0.1076” ….add “when compared to the PHA-PBMC group alone”. Do this for all sentences in this section.
- The above paragraph should end with "However, there were no significant differences between the co-culture FBS-MSC and iHPL-MSC groups". Especially since this is the point of the whole study (i.e. to compare the two).
- Page 10 Discussion, Lines 43-44 are not clear. Should it be "...and not statistically significant" (compared to co-cultures?). And what do the authors mean by "not homogeneous"?
- It is benefical to include the n # for each experiment as it is not clear how many MSC donors were used in each study. Ideally it would be good to include this in each of the figure legends.
Author Response
We have revised the manuscript following the suggestions as outlined by the reviewers. We hope that it will now be suitable for publication in your Journal. Please do not hesitate to contact us with any further questions or for further modifications.
Yours sincerely,
Katia Mareschi
As far as the comments of the Reviewers are concerned, here follows:
The review 1
This study compared bone marrow-derived MSCs that were isolated and expanded simultaneously in media containing either iHPL that was prepared using Psoralen as a pathogen inactivator or FBS. The immunophenotype of MSCs was measured. The effect of the two different MSCs were also compared in co-culture experiments with PHA-PBMCs and the different T cell subsets (naïve, memory, effector, Th1 and Th2 lymphocytes were assessed after 5 days of culture. The proliferation of PHA-PBMCs, Th1 and Th2 cells as well as the Th1 and Th2 cytokine release profile was additionally measured. They showed that iHPL-MSCs (Psoralen) had similar properties to MSCs cultured in FBS and that there were no significant differences between the two groups suggesting that iHPL-MSCs (Psoralen) could be used for GMP production of MSCs.
Major comments:
- Table I should be removed. It is inaccurate and suggests something different than what the data actually shows. Moreover, the authors state that "for each experiment a contingency table is given (Table 1- Summary of results)". This suggests that the Table is a summary of the results of the present study. The table also makes the reader believe that MSCs cultured in iHPL have a better proliferation potential which was not shown in the present study, despite being shown in reference (8). Additionally, pluripotency was not measured in the present study. And importantly, as the present study clearly showed, there were no significant differences between FBS and iHPL co-cultures. Therefore Table I is inaccurate and gives a biased impression of the results.
We have eliminated Table 1
- Methods are insufficient and need to provide more details rather than refer to other papers. Some examples. A) Specify which antibodies were used for FACS analysis. B) Define what the Physiological solution is pg. 3 and from which company, as well as where they purchased human albumin, PHA, CD45RA microbeads. C) Define which cocktail of antibodies and cytokines were used for Th1 vs. Th2 polarization, company and more details. D) At which point were the cells frozen specifically? E) Include the passage at which the cells were frozen and used for experiments as this could influence results.
As we used the same experimental study described in Mareschi e al 2016, we referred to that work for the details to avoid plagiarism. However, in the reviewed manuscript we have added the details requested by the reviewer.
- In the Methods they state that MSCs were expanded for several passages until they no longer reached confluence. They should specify how long it took until cells no longer reached confluence as this is important. Was there a difference between FBS and iHPL. Could they include this as experimental data?
We routinely isolated MSCs from the BM samples of healthy donors; those who underwent BM familiar donation collection to cure children affected by onco-hematology diseases. For the experiments performed in this case we used MSCs from our BM-MSC samples frozen in liquid nitrogen. Usually, we cultivate the MSCs until they reach senescence but for the experiments described we used BM-MSC only for the first three steps. Data about the growth of MSC cultivated in FBS and HPL have been described in Castiglia&Mareschi et al. 2014. We reviewed the material and methods section with a more precise description.
- The authors suggest that some of the cells were not compliant with ICS MSC criteria which is highly unusual and that these cells were not used. More specific details need to be provided on their selection criteria. What was their selection criteria and how specifically did they choose "compliant" cells vs. non-compliant cells?
In our work we wrote “ Only the cells which were compliant with the International Cellular Society MSC criteria [22], and which were not senescent, were frozen in FBS with 10% dimethyl sulfoxide (DMSO, Euroclone, Pero, Mi, Italy) or in physiological solution containing 5% human albumin and 10% DMSO. The cells were then thawed at the moment of the experiments in this study.”. As explained in the previous comment, we routinely isolate MSCs from BM but also from other tissues and we only froze the MSCs compliant to ICS MSC criteria. We agree with the reviewer and we confirm that on the basis of our experience and using our method, all BM-MSCs result as compliant with ICS MSC criteria.
We have reviewed the material and methods section and included a clearer description
- Were all MSCs completely negative for CD45, CD34 and CD14? The mean +/- SD should be provided. The authors should state that the CD45, CD34 and CD14 data was not shown (data not shown).
Data for the different immunophenotypes in the MSCs cultivated in FBS and iHPL have been described in Castiglia&Mareschi et al. 2014. After thawing we verified that immunophenotype remained unaltered and we reported the immunophenotypic mean with SEM of the MSCs analysed in the reviewed manuscript.
- The % positive CD146 cells should be provided especially since HPL has been shown to influence the expression of CD146 and this has been correlated with altered bone marrow MSC and platelet MSC differentiation properties (Choice of Xenogenic-Free Expansion Media Significantly Influences the Myogenic Differentiation Potential of Human Bone Marrow-Derived Mesenchymal Stromal Cells doi: 10.1016/j.jcyt.2015.11.019; Expression of CD146 on Human Placenta-Derived Mesenchymal Stromal Cells and their Osteogenic Differentiation Capacity are Modulated by Factors Contained in Platelet Lysate, doi: 10.4172/2325-9620.1000133).
We agree with the reviewer that the expression of CD146 is variable in the different cultures and we included it although it is not specific for MSC characterization as indicated in ICT MSC criteria. In a previous paper we outlined the immunophenotpe of iHPL and FB-MSCs without observing statistical differences (Castiglia &Mareschi, Cytotherapy. 2014 (doi:10.1016/j.jcyt.2013.12.008). Also here, after thawing we re-tested the immunophenotype and no statistical differences were observed. In the results section we indicate the mean values with the SEM of the analysed MSCs.
- In lines 33-35 in the Discussion on page 10, the authors state that "Proliferative and differentiative characteristics were not modified (data not shown) by cryopreservation". If the authors have the data, they should consider showing this as this would enhance the paper especially since several groups have shown differences in differentiation potential in FBS vs xenogenic-free cultures, while others have shown no differences. It would be interesting to include the data since they used a different method of HPL inactivation (Psoralen) which is not commonly used.
Thank you for the observation. We are preparing another manuscript that will report data about proliferative and differentiative characteristics at pre freezing and post thawing of MSCs cultivated in HPL and FBS. To date, we have not observe statistical significance in the daily rate between iHPL-MSCs and FBS-MSCs (t test performed on 5 MSC batches comparing pre freezing and post thawing showed respectively p=0.7308 and p=0.7159 for FBS-BM-MSCs and iHPL-BM-MSCs) but in this current manuscript our aim was to evaluate if iHPL had some effects on immunomodulant properties of BM-MSCs.
- "MSCs also produced a moderate concentration of TNF-α". Is this in response to culture? Have others shown similar results in levels of cytokines measured under similar conditions using FBS or HPL (this should be discussed)?
Thanks for your observation. I checked our raw data and noticed that there was an input error in the Elisa analysis for iHPL BM-MSCs (13.1 instead of 131). We corrected this and the graph and discussion.
- There is one citation for PHA-PBMCs but what about MSC studies? Also, what was the health and age of the PBMC and MSC donors (include in methods) as this could lead to e.g. high production of TNFa.
We modified the material and method section by inserting the requested data and adding a new MSC studies references.
- Also, in lines 21-24 in 3.3 Results, they state that “ In particular, in PHA-PBMCs memory T cells were higher than naïve cells and after co-culture with both FBS-MSCs and iHPL-MSCs it was observed that: a) This ratio reversed in favour of naïve T cells, especially naïve cytotoxic subsets”.
This is interesting. What does this indicate? The authors should speculate on this in the Discussion.
Thank you for your comments. In the discussion we have speculated on this aspect.
- Lines 39-46 Discussion paragraph need to be adapted and their data should be compared to reference (8). Moreover, this study did not "... confirm that iHPL, used as medium supplement, may be considered a more efficient additive alternative to FBS for a GMP-compliant MSC expansion" and the sentence/paragraph should be modified to reflect the data as no significant differences were shown between the two groups.
We have modified the discussion to take this into account
- English needs work. It would be beneficial to have an English speaker read through this and correct it as there are multiple areas that need modifications.
A native English speaker has reviewed the whole manuscript
Minor comments:
- Abstract – MSCs should not be capitalized and the comma should be removed after MSCs. There is a lot of incorrect use of commas.
- BM – abbreviation not defined in methods
- Line 36 Discussion, change abbreviation of FBS-MSsC to FBS-MSCs
- In lines 25-31 in 3.3 Results, the authors need to state which comparisons were done to obtain these statistical values (e.g. x compared to z group). For example, "A Two Way Anova comparison showed a significant increase of naïve cells CD45RA both helper (CD45RA+3+4+) and cytotoxic (CD45RA+3+8+) in co-cultures with PHA-PBMC 26 + iHPL-MSCs respectively with p=0.0302 and 0.1076” ….add “when compared to the PHA-PBMC group alone”. Do this for all sentences in this section.
- The above paragraph should end with "However, there were no significant differences between the co-culture FBS-MSC and iHPL-MSC groups". Especially since this is the point of the whole study (i.e. to compare the two).
- Page 10 Discussion, Lines 43-44 are not clear. Should it be "...and not statistically significant" (compared to co-cultures?). And what do the authors mean by "not homogeneous"?
- It is benefical to include the n # for each experiment as it is not clear how many MSC donors were used in each study. Ideally it would be good to include this in each of the figure legends.
All minor comments have been corrected inserting the requested modifications in the text

Reviewer 2 Report
The manuscript follows the mainstream of investigations in the field, although with some new ideas.
Major comments:
1.) Methods: HPL is commonly produced either from platelet concentrates generated by single donor platelet apheresis or by pooling platelet fractions generated from whole blood (pooled platelet concentrate) – in this case the latter approach was used. Were the platelets used outdated?
2.) Methods: Pathogen inactivation: why did the authors not use commercially available products for pathogen inactivation such as the system from Macopharma-Theraflex or Cerus-INTERCEPT? They rather used psoralen and in-house protocols - how about standardization (ref. 21)?
3.) Methods: When supernatants were analyzed for the cytokines indicated in the co-culture experiments where MSCs and PBMNCs were applied, how can the authors be sure of the cell entity producing these specific cytokines – please explain in more detail.
4.)Discussion: When the authors discuss their results on cytokine production on page 11 lines 4-27 the text is unclear and the whole paragraph needs to be re-written and clearly structured. What is the message here?
5.) Another important issue to be addressed is: When T cells are activated with PHA and cultivated for a certain period of time – e.g. 48 h – can MSC then quench the cytokine response of T cells? This would be of importance when MSCs are given as a potent cell therapeutic intervention to dampen inflammation.
6.) In conclusion – page 11 lines 39-49 – the authors state stemness potential of MSCs. I have not seen any of these results including markers of pluripotency. So please modify this statement.
7.) How are memory T cells be defined?
Minor comments:
1.) The description of the BM samples is poor. In the course of allogenic-HSCT, selected healthy family stem cell donors were used. BM is lately used more frequently due to lower risk for GvHD – in contrast to mobilized peripheral stem cells collected by apheresis that possess a higher risk for GvHD.
2.) There are multiple typing errors and the manuscript should be revised by a native speaker.
Author Response
Dear review,
We have revised the manuscript following the suggestions as outlined by the reviewers. We hope that it will now be suitable for publication in your Journal. Please do not hesitate to contact us with any further questions or for further modifications.
Yours sincerely,
Katia Mareschi
As far as the comments of the Reviewers are concerned, here follows:
The manuscript follows the mainstream of investigations in the field, although with some new ideas.
Major comments:
- Methods: HPL is commonly produced either from platelet concentrates generated by single donor platelet apheresis or by pooling platelet fractions generated from whole blood (pooled platelet concentrate) – in this case the latter approach was used. Were the platelets used outdated?
We did not use outdated Platelet Concentrates for HPL preparation.
According the blood bank's standard operating procedures, Platelet Concentrates were produced from Whole Blood-derived Buffy Coats, obtained from centrifugation and separation of Whole Blood units on the day of collection. After overnight storage at + 20 and + 24 °C, Buffy Coats were pooled and centrifuged (soft-spin) to obtain Platelet Concentrates. The details of this procedure are described in Castiglia&Mareschi et at, 2014 ad indicated in the material and method sesson (ref 21).
- Methods: Pathogen inactivation: why did the authors not use commercially available products for pathogen inactivation such as the system from Macopharma-Theraflex or Cerus-INTERCEPT? They rather used psoralen and in-house protocols - how about standardization (ref. 21)?
We do use commercially available systems for Pathogen inactivation.
It is the INTERCEPT™ Blood System that comes from Cerus Corporation that uses amotosalen HCl and illumination with ultraviolet A (UVA) light to irreversibly block the replication of DNA and RNA, thus preventing the proliferation of susceptible pathogens.
Amotosalen is one of a group of photoactive compounds called psoralens whose interaction with DNA and RNA is highly specific. Once inside a pathogen, amotosalen docks in between the nucleic acid base pairs. Upon illumination with ultraviolet A (UVA) light, an interstrand crosslink is formed ‘locking’ the DNA or RNA together so that it can no longer replicate.
Pathogen reduction technologies (PRT) performed in the blood centers are named "In House procedures" to differentiate them from the PRT used by the manufacturers of human blood and plasma derived products (solvent-detergent method).
When using PRT, blood banks should use commercially available, CE-marked technologies.
- Methods: When supernatants were analysed for the cytokines indicated in the co-culture experiments where MSCs and PBMNCs were applied, how can the authors be sure of the cell entity producing these specific cytokines – please explain in more detail.
We have better explained this in the material and methods section.
4.)Discussion: When the authors discuss their results on cytokine production on page 11 lines 4-27 the text is unclear and the whole paragraph needs to be re-written and clearly structured. What is the message here?
We have reviewed the paragraph and made the message clearer. We have also modified a further paragraph in the discussion section as requested by second reviewer
- Another important issue to be addressed is: When T cells are activated with PHA and cultivated for a certain period of time – e.g. 48 h – can MSC then quench the cytokine response of T cells? This would be of importance when MSCs are given as a potent cell therapeutic intervention to dampen inflammation.
We have reviewed the discussion and this issue has been speculated on
- In conclusion – page 11 lines 39-49 – the authors state stemness potential of MSCs. I have not seen any of these results including markers of pluripotency. So please modify this statement.
Data about the stemness potential of MSCs cultivated in iHPL and FBS have just been publicised in Castiglia &Mareschi, Cytotherapy. 2014 (doi:10.1016/j.jcyt.2013.12.008). We deleted the table 1 and modified the conclusions as suggested also by the other reviewer.
- How are memory T cells be defined?
In the paragraph 3.3 T memory cells have been defined as Cytotoxic T memory cells: CD45RO+/CD3+/CD8+ and Th memory cells: CD45RO+/CD3+/CD4+ as also reported in Mareschi et al, 2016 (ref 18).
Minor comments:
1.) The description of the BM samples is poor. In the course of allogenic-HSCT, selected healthy family stem cell donors were used. BM is lately used more frequently due to lower risk for GvHD – in contrast to mobilized peripheral stem cells collected by apheresis that possess a higher risk for GvHD.
We have added this information in detail in the material and methods section
2.) There are multiple typing errors and the manuscript should be revised by a native speaker.
A native English speaker has reviewed the whole manuscript

Round 2
Reviewer 1 Report
The authors have addressed all comments adequately.
Reviewer 2 Report
All points raised in the previous review have been adequately addressed; no futher comments.